# Beyond Length: Quantifying Long-Range Information for Long-Context LLM Pretraining Data

**Haoran Deng**[1][*][†]   **Yingyu Lin**[2][*]   **Zhenghao Lin**[3]   **Xiao Liu**[3]
**Yizhou Sun**[1]   **Yi-An Ma**[2]   **Yeyun Gong**[3][‡]

[1] University of California, Los Angeles, [2] University of California, San Diego,
[3] Microsoft Research

`{denghaoran,yzsun}@cs.ucla.edu,`
`{yil208,yianma}@ucsd.edu,`
`{xiaoliu2,zhenghaolin,yegong}@microsoft.com`

## ABSTRACT

Long-context language models unlock advanced capabilities in reasoning, code generation, and document summarization by leveraging dependencies across extended spans of text. However, a significant portion of readily available long-text data lacks meaningful long-distance dependencies; most spans can be predicted using only local context. Training on such data is inefficient, making careful data selection crucial. Therefore, we introduce LongFilter, a framework for curating training data tailored to long-context pretraining. LongFilter measures the information gain provided by extended context by contrasting model predictions under long-context versus short-context settings, thereby identifying samples where long-range dependencies are essential. Experiments with LLaMA-3-8B, extending its context length from 8K to 64K, show that LongFilter efficiently selects high-quality data and yields substantial improvements on benchmarks such as HELMET, LongBench, and RULER.

## 1 INTRODUCTION

Modern large language models have shown remarkable capabilities when processing short spans of text, but many real-world tasks, such as reasoning across documents, generating long codebases, or summarizing entire chapters require understanding and integrating information over longer contexts. To enable their long-context capabilities, language models are typically first pre-trained on short-context data and then further continually pre-trained on long-context data, which activates their long-context reasoning abilities. Although recent advances, such as RoPE modifications (Su et al., 2024) and attention interpolation (Peng et al., 2023; Ding et al., 2024), have improved efficiency and reduced training costs, the choice of training data remains a critical determinant of performance.

While existing methods improve long-context pretraining efficiency, the quality of long-context training data remains a critical factor for unlocking a model's long-context abilities. Current data engineering approaches primarily focus on sequence length, for example, by increasing the proportion of long sequences in the training set (Fu et al., 2024; Abdin et al., 2024; Yang et al., 2025) or adjusting the ratio between long and short sequences (Gao et al., 2024). However, sequence length alone fails to differentiate genuinely long-context-dependent data from lengthy sequences dominated by repetitions, independent segments, or tokens predictable from short preceding spans. As a result, a considerable portion of long sequences in commonly used corpora does not truly contain long-distance dependencies, since their next tokens can already be inferred from short contexts. Even high-quality samples may thus be more appropriate for short-context rather than long-context pretraining.

---

[*]Equal contribution.
[†]Work done during his internship at Microsoft Research.
[‡]Corresponding author.

Long-length data does not mean long-context data. Some long-length texts simply lack contextual information in their extended content. For instance, consider books and poetry collections. Individual poems, even by the same author, often lack inter-poem dependencies, and their relatively short length makes them suitable for short-context models. In contrast, textbooks are more appropriate for long-context pretraining, as chapters are tightly interconnected and understanding one chapter often requires access to preceding chapters. These examples illustrate that not all long sequences convey meaningful long-context information. Including sequences that do not require extended context can dilute the training signal and ultimately lead to performance degradation on long-context tasks.

Building on this motivation, we explored how to select training data that not only features longer length but also leverages information dependencies over longer distances. Existing long-context pre-training strategies can be viewed as a "0-to-1" step: by increasing the proportion of long sequences, the model begins to learn long-context dependencies. However, because the training loss is averaged over all tokens, sequences that do not truly depend on long-range context contribute equally to the learning signal, which is suboptimal. In this work, we take a "1-to-2" step by further increasing the proportion of sequences that genuinely require long-context understanding. This approach assigns more learning signal to tokens that depend on extended context, improving the efficiency of long-context pretraining and enabling the model to better leverage long-range dependencies.

To distinguish truly useful long-context data from merely long-length sequences, we propose Long-Filter, a data selection framework for continued pre-training. Our method is founded on a simple yet powerful principle: data is valuable for long-context training only if the long context actually helps the model make better predictions. We operationalize this insight by developing a scoring function to quantify this "information gain." The score's formulation is derived from the Kullback-Leibler (KL) divergence between a model's next-token prediction distributions conditioned on a long versus a short context, which we compute at each token position. These token-level scores are then aggregated to produce a final score for the entire sequence. A high score signifies that the extended context provides crucial information, making the sequence a high-quality candidate for training.

**Contributions**

1. This paper suggests that long-context continued pretraining should be conducted on data whose extended context provides additional information for next-token prediction.

2. We propose LongFilter, a data curation method that quantifies the information gain provided by an extended context. Using a transformer-based causal language model, LongFilter efficiently scores and selects high-quality long-context pre-training data.

3. Extensive experiments show that, without modifying the model or training setup, simply selecting training data with richer long-range information can substantially improve a language model's long-text processing ability during continued pre-training. Models trained on LongFilter-selected data achieved average gains of over 2 points on benchmarks including HELMET, LongBench, and RULER.

## 2 RELATED WORK

### 2.1 LONG-CONTEXT LANGUAGE MODEL PRETRAINING

Long-context language models have garnered significant attention within the community in recent years due to their high practical value in applications such as code generation and reasoning. A current mainstream approach involves extending the context of an existing language model with short-term context. On top of this, certain techniques have been developed to reduce the amount of training required. For example, some works employ position interpolation (Chen et al., 2023; Peng et al., 2023; Bertsch et al., 2023; Ding et al., 2024; Liu et al., 2024b; Zhang et al., 2024; Zhu et al., 2024) on RoPE (Su et al., 2024) to enable the model to better adapt to the positional encoding of extended context, or manipulating attention module (Xiong et al., 2025; Jin et al., 2024; Bertsch et al., 2023). Some of these methods have been applied in certain enterprise-level models (Liu et al., 2024a; Yang et al., 2025).

## 2.2 Data Curation and Filtering for Language Model Pretraining

The quality of data exerts a direct influence on the performance of language models. This has become a standard process for enterprise-level langauge models (Gunasekar et al., 2023; Abdin et al., 2024; Abouelenin et al., 2025). Typically, this complex process involves multiple steps, including heuristic approaches (Gao et al., 2020; Laurençon et al., 2022; Rae et al., 2021), data quality classification (Longpre et al., 2024; Wettig et al., 2024; Xie et al., 2023), domain-specific selection (Feng et al., 2022), deduplication (Borgeaud et al., 2022; Abbas et al., 2023), multilingual filtering (Wenzek et al., 2019), removing toxic content (Penedo et al., 2023; Jansen et al., 2022). These methods have achieved tremendous success in short-context model pretraining, yet few of them are specifically designed for long-context data.

## 2.3 Data Engineering for Long-Context Pretraining

Existing data engineering approaches for long-context pretraining primarily focus on the length of training data, specifically by adjusting data proportions to increase the proportion of longer-length training examples within the text corpus (Abdin et al., 2024; Yang et al., 2025). Fu et al. (2024) recommends increasing the proportion of data with longer length while maintaining domain balance. Gao et al. (2024) investigated the impact of the ratio of long-to-short data mixing and the data source on the performance of long-text pretraining. A similar idea to this paper is LongWanjuan (Liu et al., 2024c), which proposes several metrics to measure the quality of long text data. However, most of its metrics are also applicable to short texts, and the context length of the model-based filtering method used in its paper is too short (the longest window in its paper is as short as the short windows in this paper). Another related approach, LongAttn (Wu et al., 2025), uses attention scores to model long-range dependencies, but studies have shown that these attention scores do not reliably capture token importance.

## 3 Methodology

Our method, LongFilter, is designed to identify and select training data where long-range dependencies are semantically meaningful and essential for accurate token prediction. The core insight is to quantify the "information gain" provided by an extended context. We formalize this gain as the (surrogate ) Kullback-Leibler (KL) divergence between the predictive distributions of a language model given a long context versus a short one, which we compute at each token position. These token-level scores are then aggregated to produce a final score for the entire sequence.

Based on this principle, our framework follows a multi-step pipeline:

1. **Calculate Token-Level Gain:** At each token's position, we compute a score that quantifies the information gain the extended context provides for predicting that specific token.

2. **Aggregate Document Score:** We aggregate these token-level scores (e.g., by averaging) to produce a single, final score for the entire data instance.

3. **Filter and Select:** We rank all instances by this aggregate score and select a high-scoring subset for continued pre-training.

### 3.1 Problem Formulation and Notation

We operate within the standard causal language modeling framework, where the objective is to predict the next token $x_t$ given a preceding context $x_{<t}$.

Let a sequence of tokens be denoted by $X = (x_1, x_2, \ldots, x_N)$. For any given token $x_t$ in the sequence, we define two distinct context windows:

- **Short Context** ($S$): The sequence of $\ell_{\text{Short}}$ tokens immediately preceding $x_t$. Formally, $S(t) = (x_{t-\ell_{\text{Short}}}, \ldots, x_{t-1})$.

- **Long Context** ($L$): The sequence of $\ell_{\text{Long}}$ tokens immediately preceding $x_t$, where $\ell_{\text{Long}} > \ell_{\text{Short}}$. Formally, $L(t) = (x_{t-\ell_{\text{Long}}}, \ldots, x_{t-1})$.

The **extended context**, denoted $E$, is the portion of the long context that precedes the short context, i.e., $E(t) = (x_{t-l_l}, \ldots, x_{t-l_s})$. The long context is therefore the concatenation of the extended and short contexts, $L = E \circ S$. See Figure 1.

| | | |
|---|---|---|
| **S**hort context | I hate this ? | |
| **L**ong context | The plot was a mess and the acting was terrible. I hate this ? | song |
| **E**xtended context | The plot was a mess and the acting was terrible. | movie |
| Next **T**oken | ? | thing |

Figure 1: An illustration of the token-level long-context information gain. Given only the Short Context ($S$) "I hate this", the predictive distribution for the next token has high entropy, as many words ('song', 'thing', 'movie') are plausible. The Extended Context ($E$), "The plot was a mess...", provides critical information that reduces this entropy, concentrating the probability on "movie".

Given a pre-trained language model $M$, we can obtain two conditional probability distributions for the next token:

$$P_S(\cdot) = P_M(\cdot \mid S(t)) \quad \text{and} \quad P_L(\cdot) = P_M(\cdot \mid L(t))$$

The central question LongFilter addresses is:

*How can we quantify the additional information that the extended context $E$ provides for predicting $x_t$ beyond what is already available in the short context $S$?*

### 3.2 Information-Theoretic Formulation of Contextual Gain

The ideal theoretical tool to answer our question is *Conditional Mutual Information (CMI)*. The CMI $I(T; E \mid S)$ measures the reduction in uncertainty about a target variable $T$ (the next token) after observing an extended context $E$, given that a short context $S$ is already known (Cover & Thomas, 2006).

The CMI can be expressed in two well-known, equivalent forms. The first defines CMI as the reduction in conditional entropy:

$$I(T; E \mid S) = H(T \mid S) - H(T \mid S, E) \tag{1}$$

where $H(\cdot \mid \cdot)$ is the conditional entropy. A second, equivalent formulation expresses the CMI as the expected Kullback-Leibler (KL) divergence between the predictive distributions with and without the extended context:

$$I(T; E \mid S) = \mathbb{E}_{p(s,e)} \left[ D_{KL}\big(p(T \mid S = s, E = e) \,\|\, p(T \mid S = s)\big) \right] \tag{2}$$

This second form is particularly insightful, as it frames the information gain as the expected "distance" between the posterior belief $p(T \mid S, E)$ and the prior belief $p(T \mid S)$. For completeness, we derive the equivalence of these two definitions in Appendix C.

For a given context instance $(e^*, s^*)$, to evaluate the effect of the extended context $e^*$ on the next token $T$ prediction, we consider the one sample estimate of the above CMI:

$$\hat{I}(T; E = e^* \mid S = s^*) = D_{KL}\big(p(T \mid S = s^*, E = e^*) \,\|\, p(T \mid S = s^*)\big) \tag{3}$$

### 3.3 A Practical Scoring Function for Contextual Gain

Expanding the KL divergence by its definition in equation 3, we have

$$D_{KL}\big(p(T \mid S = s^*, E = e^*) \,\|\, p(T \mid S = s^*)\big) = \sum_{t \in \mathcal{V}} p(t \mid s^*, e^*) \log \frac{p(t \mid s^*, e^*)}{p(t \mid s^*)}. \tag{4}$$

This formula has two drawbacks: does not leverage the ground-truth information of $T = t^*$, i.e., the value of $D_{KL}\big(p(T \mid S = s^*, E = e^*) \,\|\, p(T \mid S = s^*)\big)$ does not depend on $t^*$ and it requires a costly summation over the entire vocabulary $\mathcal{V}$. To create a practical score for a single ground-truth instance $(t^*, s^*, e^*)$, we focus on the term corresponding to $t^*$, which yields a surrogate for KL divergence:

$$\text{score}(t^*, s^*, e^*) = p(T = t^* \mid E = e^*, S = s^*) \log \frac{p(T = t^* \mid E = e^*, S = s^*)}{p(T = t^* \mid S = s^*)} \tag{5}$$

This score (Eq. 5) provides our fundamental "token-level score". It can be interpreted as the gain for predicting the specific target $t^*$ that is contributed by the extended context $e^*$, given that $s^*$ was already observed.

To evaluate an entire data instance $X^* = (x_1^*, \ldots, x_N^*)$—as outlined in our methodology—we aggregate these individual scores. We define the final **LongFilter score** as the average of the per-token scores across the sequence:

$$
\begin{aligned}
\text{Score}(X^*) &= \frac{1}{N} \sum_{i=1}^{N} \text{score}(x_{i-\ell_{\text{Long}}:i-\ell_{\text{Short}}-1}^*, x_{i-\ell_{\text{Short}}:i-1}^*, x_i^*) \\
&= \frac{1}{N} \sum_{i=1}^{N} p(x_i^* \mid x_{i-\ell_{\text{Long}}:i-1}^*) \log \frac{p(x_i^* \mid x_{i-\ell_{\text{Long}}:i-1}^*)}{p(x_i^* \mid x_{i-\ell_{\text{Short}}:i-1}^*)}
\end{aligned}
\tag{6}
$$

For a more practical perspective, we can reformulate the LongFilter score in terms of the standard per-token cross-entropy loss, which is equivalent to the negative log-likelihood. Let $\mathcal{L}^{\text{long}}$ and $\mathcal{L}^{\text{short}}$ be the losses for predicting the ground-truth token $x_i^*$ given the long and short contexts, respectively:

$$
\begin{aligned}
\mathcal{L}_i^{\text{long}} &= H_c\left(\mathbf{1}\{x_i = x_i^*\}, p(\cdot \mid x_{i-\ell_{\text{Long}}:i-1}^*)\right) = -\log p(x_i^* \mid x_{i-\ell_{\text{Long}}:i-1}^*), \\
\mathcal{L}_i^{\text{short}} &= H_c\left(\mathbf{1}\{x_i = x_i^*\}, p(\cdot \mid x_{i-\ell_{\text{Short}}:i-1}^*)\right) = -\log p(x_i^* \mid x_{i-\ell_{\text{Short}}:i-1}^*),
\end{aligned}
$$

where $H_c(p, q) = -\mathbb{E}_p \log q$ denotes the cross-entropy of the distribution $q$ relative to $p$.

Then we have

$$
\text{Score}(X^*) = \frac{1}{N} \sum_{i=1}^{N} \exp(-\mathcal{L}_i^{\text{long}})(\mathcal{L}_i^{\text{short}} - \mathcal{L}_i^{\text{long}}).
\tag{7}
$$

This loss-based view offers a clear interpretation: the score gives preference to examples where the reduction in prediction loss from using a longer context (the term $\mathcal{L}^{\text{short}} - \mathcal{L}^{\text{long}}$) is large. This loss reduction is then weighted by the model's confidence on the token given the full context $(\exp(-\mathcal{L}^{\text{long}}) = p(x_i^* \mid x_{i-\ell_{\text{Long}}:i-1}^*))$, ensuring that the gains are on tokens the model considers plausible.

## 3.4 LONGFILTER

The framework of LongFilter is shown in Figure 2. LongFilter utilizes a pre-trained causal language model to estimate the distribution of the next token across varying context lengths. LongFilter consists of three steps: Long-context Modeling, Short-Context Modeling, and LongFilter Scoring.

**Long-Context Modeling.** For an input sequence, we compute the probability distribution obtained by predicting the next token for each position based on its prefix context, in a manner analogous to the training stage of causal language model. The process of getting the next token distribution in the long context utilizes the prefix context from all positions.

**Short-Context Modeling.** For predicting the distribution of the next token in a short context, the LongFilter first segments the entire text into shorter chunks. Each short chunk is then fed into a pre-trained causal language model, thereby constraining the context of the predicted output to the boundaries of the short chunk. To avoid predicting tokens with insufficient context at the beginning of each short chunk, we chose to introduce overlap between different chunks during segmentation.

**LongFilter Scoring.** After obtaining the probabilities for predicting the next token from both short and long contexts, the final score is calculated using Equation 7. All scores are sorted, and a portion of the higher-scoring entries are selected as the chosen data.

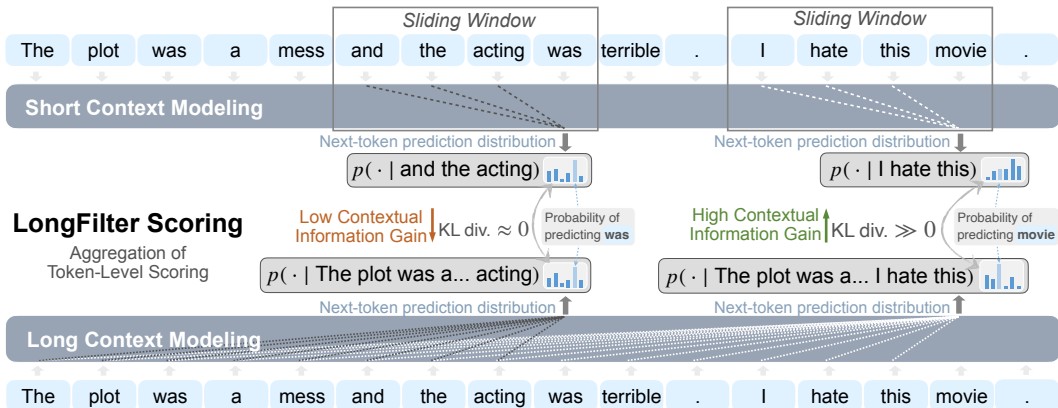

Figure 2: Workflow of LongFilter. The **Upper part** computes the next-token probability distribution using a short-context sliding window (shown as 4 tokens for illustration, though our experiments use 4K), while the **Lower part** computes it using the full long context. LongFilter then scores the information gain (**Middle part**) by calculating a token-level surrogate KL divergence between these two distributions. This gain is low for locally predictable tokens (such as 'was'), but high for tokens that require extended context (such as 'movie'). Finally, these token-level scores are aggregated to produce a single score for the entire data instance.

## 4 EXPERIMENTS

### 4.1 SETUP

We conduct experiments by continually pre-training LLaMA-3-8B (Dubey et al., 2024), which has an initial effective context length of 8K, on different datasets to extend its context length to 64K. For each dataset, we pack the tokenized text into sequences of 64K tokens, score each sample with LongFilter. The model is then continually pre-trained on these filtered samples, and its performance is evaluated on long-context benchmarks to assess the effectiveness of the data selection process.

**Datasets.** We use SlimPajama-627B (Soboleva et al., 2023) as the primary source for long-context pretraining. LongFilter is applied to extract high-quality long-text samples from this dataset. SlimPajama has been widely adopted in recent long-context data engineering works (e.g., Gao et al. (2024), Fu et al. (2024)).

Specifically, we select three corpora from SlimPajama, that is ArXiv, Books, and CommonCrawl, for our experiments. We similarly categorized each corpus by length, selecting thresholds of 16K, 64K, and 32K for long and short texts in ArXiv, Book, and CommonCrawl, respectively. After applying these thresholds, the volume of data classified as long texts was approximately 19 billion tokens. We constructed the model training dataset with 80% long texts and 20% short texts. Data selection was applied exclusively to the long text portion.

Table 1: Statistics of long context tokens.

| Name | #Tokens |
|---|---|
| SlimPajama-Book | 19,535,822,848 |
| SlimPajama-Arxiv | 19,489,295,000 |
| SlimPajama-CommonCrawl | 19,284,099,072 |

**Model and Training Configuration.** For training, we adopt the same configurations as Pro-Long (Gao et al., 2024) when scaling LLaMA-3-8B from an 8K to a 64K context, including optimizer, learning rate, and RoPE base frequency. The only difference lies in our choice of training data, which is guided by LongFilter-based selection. Apart from increasing the RoPE base frequency from $5 \times 10^5$ to $8 \times 10^6$, we made no further modifications to the model in our experiments. Drawing on configurations from previous studies on long text training (Fu et al., 2024), we set the batch size to 4M tokens and trained for 1,000 steps, processing a total of 4B tokens.

**Baselines.** We first compared our model with ProLong (Gao et al., 2024), but unified the training data to three corpora from the SlimPajama dataset and adopted the same short-to-long ratio. For a fair comparison, we did not use ProLong's ShortMix dataset. ProLong's training data was sampled from all training data, while LongFilter's training data was sampled from the selected data. We did not exclude the selected data from ProLong's training set, meaning ProLong and LongFilter share a portion of high-quality long-context training data. We also compared our approach with LongWanjuan (Liu et al., 2024c), conducting comparative experiments using their best-performing aggregated and holistic data ratio of 1:1 as specified in their paper.

**Setting of LongFilter.** We set the short context window to 4K and the long context window to 64K, using the Llama-3.1-8B model (which supports 128K contexts) for scoring. We sorted the scores and selected the top 20% of data as the final training dataset of LongFilter. We run LongFilter on 32 NVIDIA H100 GPUs, enabling each corpus to complete all scoring within a single day.

## 4.2 EVALUATION ON RECALL (NEEDLE-IN-A-HAYSTACK) TASKS

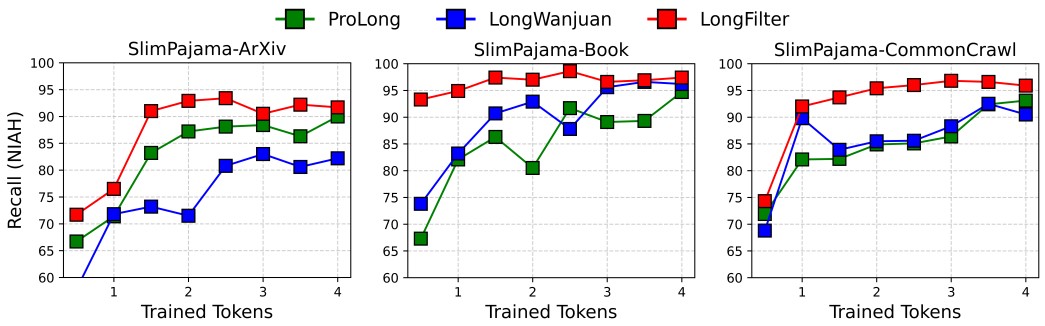

Figure 3: Performance on Recall tasks (Needle-in-a-Haystack) w.r.t trained tokens.

We first report the performance of different data strategies on a series of Recall tasks. This series of tasks has also been referred to as Needle-in-a-Haystack (NIAH) (Kamradt, 2023). This type of tasks directly tests a model's ability to utilize information from any position, often serving as one of the most important metrics for evaluating a model's performance on long text.

Specifically, we reported on the SubEM of Recall task within the HELMET benchmark (Yen et al., 2025), which encompasses four distinct NIAH tasks: JsonKV, Needle retrieval with multiple keys, UUID retrieval with multiple keys, and value retrieval with multiple keys. Experimental results are shown in Figure 3.

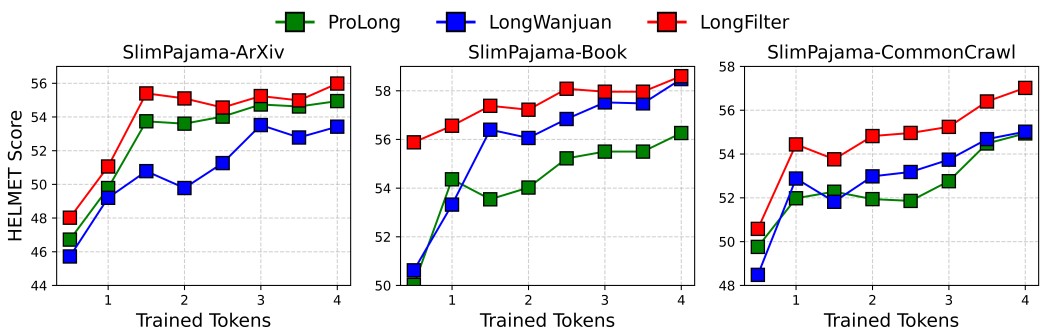

Figure 4: Performance on HELMET with respect to trained tokens.

Across the three experimental settings with different type of training data, we consistently observe that LongFilter achieves the best overall performance and exhibits the most stable improvement as the training scale increases. For example, in all three groups, LongFilter rapidly surpasses both ProLong and LongWanjuan at small scales (0.5B–1B) and maintains a clear advantage when scaling up to 4B, reaching performance above 90 in every case. This indicates that filtering with LongFilter can effectively enhance data quality and maximize the benefits of scaling.

## 4.3 EVALUATION ON LONG-CONTEXT BENCHMARKS

Table 2: Experimental result on LongBench.

| Dataset & Model | SingleQA | MultiQA | Summ | ICL | Synthetic | Code | Overall |
|---|---|---|---|---|---|---|---|
| **Arxiv** | | | | | | | |
| ProLong | 25.26 | 16.24 | 24.83 | 63.04 | 51.76 | 68.69 | 38.58 |
| LongWanjuan | 28.43 | 18.25 | 25.09 | 62.85 | 49.92 | 69.22 | 39.36 |
| LongFilter | 28.32 | 17.99 | 24.48 | 63.15 | 51.41 | 70.00 | **39.52** |
| **Book** | | | | | | | |
| ProLong | 21.25 | 15.06 | 23.64 | 62.60 | 52.41 | 69.76 | 37.47 |
| LongWanjuan | 22.33 | 14.84 | 22.75 | 63.46 | 52.62 | 70.05 | 37.69 |
| LongFilter | 26.58 | 20.14 | 24.36 | 62.69 | 53.58 | 70.08 | **39.81** |
| **CC** | | | | | | | |
| ProLong | 21.95 | 16.83 | 25.03 | 63.87 | 52.18 | 69.33 | 38.37 |
| LongWanjuan | 33.64 | 18.58 | 25.18 | 61.95 | 47.87 | 69.79 | 40.02 |
| LongFilter | 30.54 | 17.21 | 25.58 | 62.84 | 57.02 | 69.11 | **40.66** |

Table 3: Experimental result on RULER.

| Dataset & Model | NIAH Single | NIAH MultiKey | NIAH MultiValue | NIAH MultiQuery | Other | Overall |
|---|---|---|---|---|---|---|
| **Arxiv** | | | | | | |
| ProLong | 77.90 | 85.40 | 89.35 | 85.73 | 47.12 | 69.28 |
| LongWanjuan | 83.8 | 80.57 | 83.16 | 85.04 | 49.15 | 69.69 |
| LongFilter | 78.68 | 86.20 | 89.76 | 86.92 | 48.07 | **70.13** |
| **Book** | | | | | | |
| ProLong | 93.83 | 83.80 | 92.76 | 95.08 | 46.56 | 73.35 |
| LongWanjuan | 90.08 | 91.03 | 91.38 | 82.75 | 48.23 | 73.74 |
| LongFilter | 95.33 | 97.87 | 93.15 | 80.10 | 54.71 | **78.95** |
| **CommonCrawl** | | | | | | |
| ProLong | 91.31 | 86.58 | 94.32 | 77.87 | 47.54 | 72.59 |
| LongWanjuan | 90.85 | 84.65 | 89.76 | 80.49 | 41.25 | 74.08 |
| LongFilter | 92.58 | 94.50 | 94.80 | 77.80 | 31.71 | **75.37** |

To validate whether the data selection strategy of LongFilter is beneficial for training long-text language models, we evaluate the continually pre-trained models on 3 widely used long-context benchmarks: HELMET (Yen et al., 2025), LongBench (Bai et al., 2024), and RULER (Hsieh et al., 2024). Since RULER and LongBench require language models to comprehend instructions, we SFT models with 1B data using UltraChat dataset (with settings consistent with Gao et al. (2024)).

The final reported score is the average of all non-model-based evaluation metrics across all tasks in HELMET, encompassing five tasks: Recall, RAG, Re-rank, ICL, and QA. We report the overall performance on HELMET benchmark with respect to the number of trained tokens in Figure 4.

According to the experimental results, the quality of long-text training data undergoes a clear, significant, and sustained improvement after LongFilter's data selection. LongFilter significantly improves training efficiency. Compared to unfiltered data, length extension training with 1.5B filtered tokens

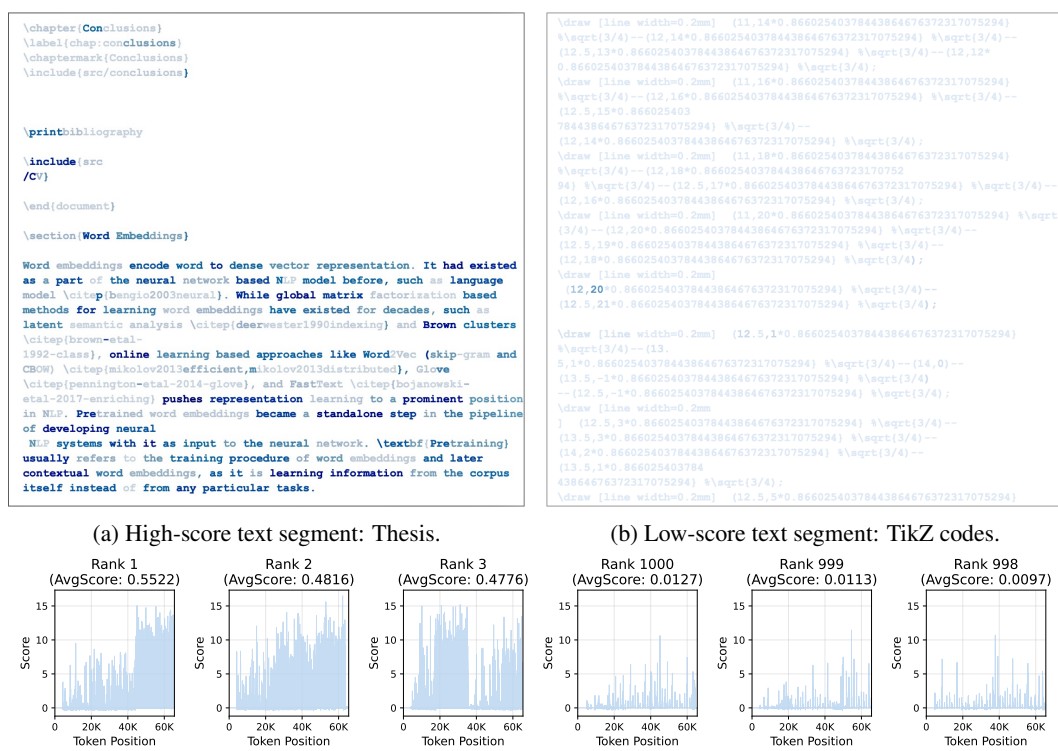

(a) High-score text segment: Thesis.  (b) Low-score text segment: TikZ codes.

(c) Context Score at each token position for documents of varying ranks.

Figure 5: Token-level context score analysis on a subset of the processed SlimPajama-Arxiv dataset. In the top examples, the color of each token is determined by its score: the darker the color, the higher the score. (a) A high-scoring segment of well-formed academic prose from a PhD thesis. (b) A low-scoring segment containing non-prose LaTeX TikZ drawing commands. (c) Context Scores across the full token sequence for documents of the top three ranks and the bottom three ranks.

already achieves performance comparable to training on 3-4B tokens, indicating that only about half the data is required to reach the same level of effectiveness.

On LongBench, the three methods demonstrate different trade-offs across datasets. LongFilter consistently achieves the highest overall scores, showing its robustness across diverse domains. In particular, it yields notable gains in Synthetic tasks (e.g., 57.02 on CC) and maintains competitive performance in Code, where both categories heavily rely on the model's ability to leverage information from arbitrary positions within the context. This suggests that LongFilter effectively improves the long-range information of training data, thereby benefiting tasks needs long-range dependency.

On the RULER benchmark, LongFilter consistently achieves the highest overall scores across all three datasets. Its advantage is particularly pronounced in structured data tasks, such as MultiKey, MultiValue, and MultiQuery, where careful filtering likely enhances the model's ability to capture long-range information.

Overall, these results reinforce the pattern observed in LongBench: data quality and filtering (Long-Filter) provide more consistent and robust improvements than original data and LongWanjuan.

## 4.4 CASE STUDY: TOKEN-LEVEL ANALYSIS

In this case study, we analyze a subset of the processed SlimPajama-Arxiv dataset containing 1000 samples, each a sequence of 65536 tokens. The analysis is presented in Figure 5. The token-level score is visualized by color intensity in Figure 5a (prose) and Figure 5b (code), where darker text indicates a higher score. These results support the intuition that repetitive content like TikZ code, which lacks long-range semantic structure, receives low scores. Figure 5c shows the token-level scores for the top-three and bottom-three ranked documents. The abrupt score jumps seen in plots

of Rank 1 and 3 are artifacts created by concatenating multiple `.tex` files from a single arXiv submission during data preprocessing.

## 5 CONCLUSION

This paper proposes a data filtering framework tailored for pretraining long-context language models. Unlike short-context language models, long-context models require leveraging semantic information from longer range of positions. Based on intuition, we recommend that long-context language models should be trained on data where this additional length provides information for the next word prediction.

We formalize this process as identifying training data where additional context yields higher conditional mutual information for predicting the next token. Based on this formulation, we develop a scoring function that estimates the informational gain of context using a trained language model. To apply this method to practical data filtering, we design a model called LongFilter to score the informational value of additional context in long training data, recommending training on data with higher scores.

Sufficient experimental resutls demonstrates the effectiveness of LongFilter. We achieve sustained and significant improvements in long-text capabilities for long-text models solely through data filtering. After expanding the Llama-3-8B model from 8K to 64K context, experiments on benchmarks like HELMET, LongBench, and RULER demonstrate that this simple yet effective method yields up to a 10% accuracy gain on recall tasks when training on 1B tokens.

## ACKNOWLEDGEMENTS

Haoran Deng and Yizhou Sun were partially supported by NSF 2211557, NSF 2119643, NSF 2303037, NSF 2312501, NSF 2531008, SRC JUMP 2.0 Center, Amazon Research Awards, and Snapchat Gifts.

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

## A  REPRODUCIBILITY

All code and scripts used in this work are publicly available at: https://github.com/HaoranDeng/LongFilter.

This repository contains all necessary files for data preprocessing, model training, evaluation, and visualization.

## B  USE OF LLM

During the preparation of this paper, a large language model (LLM) was used solely for the following purposes:

- Sentence-level polishing of grammar and wording.
- Translation.

The LLM was not used to generate original content, draft sections of the paper, or make any scientific claims. The authors take full responsibility for all content in the submission.

## C  EQUIVALENCE OF TWO DEFINITIONS OF CONDITIONAL MUTUAL INFORMATION

The equivalence of Eq. equation 1 and Eq. equation 2 can be shown by expanding the definition of entropy:

$$
\begin{aligned}
I(T; E \mid S) &= H(T \mid S) - H(T \mid S, E) \\
&= \left( -\sum_{s,t} p(s,t) \log p(t \mid s) \right) - \left( -\sum_{s,e,t} p(s,e,t) \log p(t \mid s,e) \right) \\
&= -\sum_{s,e,t} p(s,e,t) \log p(t \mid s) + \sum_{s,e,t} p(s,e,t) \log p(t \mid s,e) \\
&= \sum_{s,e,t} p(s,e,t) \left( \log p(t \mid s,e) - \log p(t \mid s) \right) \\
&= \sum_{s,e,t} p(s,e,t) \log \frac{p(t \mid s,e)}{p(t \mid s)} \\
&= \sum_{s,e} p(s,e) \sum_t p(t \mid s,e) \log \frac{p(t \mid s,e)}{p(t \mid s)} \\
&= \sum_{s,e} p(s,e) D_{KL}\big(p(T \mid S = s, E = e) \,\|\, p(T \mid S = s)\big) \\
&= \mathbb{E}_{p(s,e)} \big[ D_{KL}\big(p(T \mid S, E) \,\|\, p(T \mid S)\big) \big].
\end{aligned}
\tag{8}
$$

