# OpenReview forum: "Beyond Length: Quantifying Long-Range Information for Long-Context LLM Pretraining Data"
_ICLR.cc/2026/Conference — ICLR 2026 Poster_

### Official Review · Reviewer_GcpK · 2025-11-01

**Soundness:** 3
**Presentation:** 3
**Contribution:** 3
**Rating:** 6
**Confidence:** 4

**Summary:**

This paper challenges the prevailing data engineering practice for long-context LLMs, which primarily relies on sequence length as a proxy for quality. The authors argue that many long sequences do not genuinely require long-range understanding, and training on them is inefficient. They introduce LongFilter, a data curation framework designed to select training data based on its "long-range information" content.

The core of LongFilter is a scoring function derived from information theory. It quantifies the information gain provided by an extended context by measuring the KL divergence between a model's next-token predictions under a long context versus a short context. A practical and efficient score is developed based on this principle, which prioritizes sequences where access to a longer context significantly reduces the model's prediction loss. The authors conduct experiments by continually pre-training LLaMA-3-8B to extend its context from 8K to 64K. The results show that training on data selected by LongFilter leads to substantial performance improvements on a suite of long-context benchmarks compared to baseline data selection strategies.

**Strengths:**

1. The central thesis—that data for long-context training should be selected based on its actual need for long-range information—is both intuitive and powerful.
2. LongFilter is elegantly derived from the information-theoretic concept of Conditional Mutual Information, lending it strong theoretical credibility.
3. The experimental results are compelling, showing consistent and significant improvements across multiple models and benchmarks. The ability to achieve the performance of 4B tokens of training with only 1.5B tokens of filtered data is a strong testament to the method's efficiency.
4. The proposed filtering method, while computationally intensive to run, is conceptually simple and does not require modifying the model or training process. It can be applied as a preprocessing step to any long-text dataset.
5. The token-level case study provides insightful qualitative analysis, visually demonstrating that the filter correctly identifies semantically coherent prose as high-value and repetitive or unstructured content as low-value.

**Weaknesses:**

1. The primary drawback of LongFilter is its high computational cost. The paper notes that scoring a large corpus required 32 H100 GPUs for a day. This is a significant barrier for researchers with limited resources. A discussion on more efficient approximations of the score would make the work more accessible.
2. The quality of the selected data is contingent on the capabilities of the model used for scoring. The paper uses a very strong model (LLaMA-3.1-8B). It is unclear how performance would be affected if a smaller or less capable model were used for scoring.
3. The choice of l_short (4K) and l_long (64K) seems somewhat arbitrary. The effectiveness of the filter might be sensitive to these hyperparameters, but this sensitivity is not explored.

**Questions:**

The final scoring function (Eq. 7) weights the loss difference (L_short - L_long) by exp(-L_long). This means that tokens on which the model is very uncertain even with long context (i.e., high L_long) will receive a very low score, regardless of the information gain. Did you consider alternative formulations, such as using the raw loss difference L_short - L_long? What is the justification for this specific weighting scheme?

---

> ### Author Response · Authors · 2025-11-29
>
> > Computational Cost & other models
> We added a new ablation study in Appendix D.1, where we utilize a 0.6B parameter model for data filtering. The results indicate that the data selected by this small model still yields performance gains when pre-training larger models. While we recommend using the strongest available model to maximize performance, this finding confirms that our method remains viable even under constrained computational budgets.
>
> Furthermore, data selection is a one-time investment. While the training stage is often repeated multiple times (e.g., for hyperparameter tuning, varying seeds, or architectural sweeps), the filtered dataset can be reused across all these runs. Therefore, the cost of selection is effectively amortized over the lifecycle of the project. Moreover, the scoring process requires only forward passes (no backpropagation), making it significantly faster and less memory-intensive than training steps.
>
>
> > Choice of Context Window
>
> We chose 64K because we aim to extend the model to a 64K context window, which represents the range of long-distance information the model is expected to use. The 4K context length is widely adopted as the pre-training length for many models, and we believe the model already has a strong capability to capture information within this range. We recommend that users set these two values to the initial context length and the target extended length when performing context-length extension.
>
> > Alternative formulations, such as using the raw loss difference L_short - L_long? What is the justification for this specific weighting scheme?
>
> We thank the reviewer for this insightful question regarding the weighting term. We indeed considered alternative formulations, including the raw loss difference ($L_ { {short}} - L_ { {long}}$). We selected the weighted formulation for two primary reasons:
>
> 1. Theoretical Alignment with KL Divergence: The raw loss difference represents the log-likelihood ratio ($\log \frac{P}{Q}$). However, the KL divergence—which measures the expected information gain—is defined as $E_P[log \frac{P}{Q}]$. By weighting the raw difference by $exp(-L_ { {long}}) \approx P(x)$, our score becomes a proxy for the term-wise contribution to the expected KL divergence.
>
> 2. Noise Suppression: Regarding the reviewer's concern about uncertain tokens: tokens with high $L_ { {long}}$ typically reside in the long tail of the distribution. In these regions, the raw log-likelihood ratio is often numerically unstable (high variance) despite the token having negligible probability mass. The weighting term effectively suppresses this "tail noise," ensuring the score focuses on tokens that contribute meaningfully to the divergence.
>
> We have added a new Section (Appendix E) and Figure 8 to the paper. We empirically compare the raw and weighted scores against the KL divergence. The results demonstrate that the weighted formulation provides a significantly higher fidelity approximation (lower error) to the information gain.

---

### Official Review · Reviewer_bFEo · 2025-11-01

**Soundness:** 3
**Presentation:** 3
**Contribution:** 3
**Rating:** 6
**Confidence:** 4

**Summary:**

This work focuses a key challenge in training long-context language models (LLMs): much of the available long-text data does not actually require extended context to be modeled effectively.
To tackle this, the authors propose LongFilter, a framework that quantifies the information gain from using longer contexts by comparing model predictions under short and long context settings.
This allows them to curate pretraining datasets where genuine long-range dependencies are present.
They demonstrate that applying LongFilter during pretraining, specifically when extending LLaMA-3-8B’s context length from 8K to 64K, leads to substantial improvements on several benchmarks (HELMET, LongBench, RULER).
The paper also provides analyses showing which types of text segments most benefit from long-context modeling.

**Strengths:**

1. This work addresses an important but often overlooked issue in scaling up LLMs’ context windows, i.e., not all “long” data is useful for learning true long-range dependencies.
2. The authors propose a clear and interpretable metric (information gain) for identifying valuable training samples.
3. The paper provides an additional analysis on what kinds of text benefit most from extended contexts, informing future research and practical data selection.

**Weaknesses:**

1. All the results and analysis are made on one model (i.e., LLaMA-3-8B). It could be better to show the effectiveness on more backbone models.
2. Beyond the long-context understanding tasks, to further demonstrate the improvements on processing long-range information , it could be better to show the performance on long generation tasks, such as long reasoning and long writing tasks.

**Questions:**

See weakness.

---

> ### Author Response · Authors · 2025-11-29
>
> Thank you for your review and the valuable comments.
>
> > Other models
>
> We added a new ablation study in Appendix D.1, where we utilize a 0.6B parameter model for data filtering. The results indicate that the data selected by this small model still yields performance gains when pre-training larger models. While we recommend using the strongest available model to maximize performance, this finding confirms that our method remains viable even under constrained computational budgets.
>
> > Additional Tasks, such as long reasoning and long writing tasks
>
> We believe these tasks are not suitable benchmarks for evaluating this work because they depend heavily on post-training. LongFilter is a data selection method specifically designed for the **pre-training** stage.  To the best of our knowledge, most prior work on context extension[1], long-text data engineering[2], and even commercial LLMs[3] typically do not include this component at this stage of the pipeline.
>
> [1] Peng, B., Quesnelle, J., Fan, H., & Shippole, E. YaRN: Efficient Context Window Extension of Large Language Models. In The Twelfth International Conference on Learning Representations.
>
> [2] Fu, Y., Panda, R., Niu, X., Yue, X., Hajishirzi, H., Kim, Y., & Peng, H. Data Engineering for Scaling Language Models to 128K Context. In Forty-first International Conference on Machine Learning.
>
> [3] Liu, A., Feng, B., Xue, B., Wang, B., Wu, B., Lu, C., ... & Piao, Y. (2024). Deepseek-v3 technical report. arXiv preprint arXiv:2412.19437.

---

### Official Review · Reviewer_hWVa · 2025-11-01

**Soundness:** 3
**Presentation:** 4
**Contribution:** 3
**Rating:** 6
**Confidence:** 3

**Summary:**

Much readily available long-text data does not genuinely require extended context, as most spans can be predicted using only short-range context. Not all long sequences provide meaningful long-context information, and some long sequences can dilute the training signal. To solve this, the authors propose LongFilter, a framework for curating training data tailored to long-context pretraining. The core idea of LongFilter is to quantify the "information gain" provided by an extended context. The method uses a scoring function derived from the Kullback-Leibler (KL) divergence between the next-token prediction distribution given the long context versus the short context. The score is formulated as the reduction in prediction loss when using the long context compared to the short context, weighted by the model’s confidence on the token given the full context.A high score signifies that the extended context is crucial for making accurate predictions.
The authors conducted experiments by extending the Llama-3.1-8B model from an 8K context length to 64K. They filtered data from the SlimPajama dataset, selecting the top 20% of high-scoring data for training. Models trained on LongFilter-selected data showed "substantial improvements" on benchmarks like HELMET, LongBench, and RULER, achieving average gains of over 2 points. Meanwhile, LongFilter significantly improves training efficiency. Experiments showed that training with 1.5B filtered tokens achieved performance comparable to training on 3-4B unfiltered tokens.

**Strengths:**

The paper show the fact that sequence length alone is an insufficient proxy for data quality. Much long-text data does not genuinely require long-range dependencies, which can dilute the training signal.

The proposed method, LongFilter, quantifies the informational value of extended context, which is a significant contribution for LLM pretraining.

The experiments are well-founded. The experiments use Llama-3.1-8B, extend it to a significant context length (8K to 64K) , and train on a large-scale, standard dataset SlimPajama. The method is also compared against relevant baselines like ProLong and LongWanjuan. The results show substantial improvements across a suite of demanding long-context benchmarks, including HELMET, LongBench, and RULER, and demonstrate that training on 1.5B filtered tokens can achieve performance comparable to training on 3-4B unfiltered tokens.

**Weaknesses:**

The paper highlights the training efficiency gains, but the data curation step itself appears to have a high computational cost. The authors report that scoring each corpus required 32 NVIDIA H100 GPUs for a single day.

The paper uses the Llama-3.1-8B model as the scoring model to conduct experiments and achieve favorable results. However, the paper does not test other models for generating the scores, which suggests a lack of generalizability regarding the choice of the scoring model in the experiments.

**Questions:**

The Llama-3.1-8B model was used for scoring. Have you experimented with using other models for this step? How does the choice of the scoring model impact the quality of the selected data and the performance of the final trained model?

---

> ### Author Response · Authors · 2025-11-29
>
> Thank you for your review and the valuable comments.
>
> > Computational Cost & Other model
>
> We added a new ablation study in Appendix D.1, where we utilize a 0.6B parameter model for data filtering. The results indicate that the data selected by this small model still yields performance gains when pre-training larger models. While we recommend using the strongest available model to maximize performance, this finding confirms that our method remains viable even under constrained computational budgets.
>
> Furthermore, data selection is a one-time investment. While the training stage is often repeated multiple times (e.g., for hyperparameter tuning, varying seeds, or architectural sweeps), the filtered dataset can be reused across all these runs. Therefore, the cost of selection is effectively amortized over the lifecycle of the project. Moreover, the scoring process requires only forward passes (no backpropagation), making it significantly faster and less memory-intensive than training steps.

---

### Official Review · Reviewer_xifA · 2025-11-04

**Soundness:** 3
**Presentation:** 2
**Contribution:** 2
**Rating:** 4
**Confidence:** 4

**Summary:**

This paper introduces LongFilter, a data curation framework that selects long-context pretraining data based on the information gain provided by extended context, measured via KL divergence between next-token prediction distributions under short (e.g., 4K) and long (e.g., 64K) contexts. The core insight—that sequence length alone is insufficient to ensure meaningful long-range dependencies—is well-motivated and validated through experiments on LLaMA-3-8B extended from 8K to 64K context. Results on HELMET, LongBench, and RULER show consistent gains (e.g., +2+ average points), with particularly strong improvements in recall (Needle-in-a-Haystack) tasks.

**Strengths:**

* Novel and principled metric: The use of conditional mutual information (via KL divergence) to quantify long-range dependency is theoretically grounded and practically effective.

* Strong empirical validation: Experiments across multiple benchmarks and data domains (ArXiv, Books, CommonCrawl) demonstrate robustness. LongFilter outperforms length-based baselines (ProLong) and a recent alternative (LongWanjuan).

* High practical impact: Achieves comparable performance with half the training tokens, significantly improving data efficiency.

* Transparent methodology: The scoring function (Eq. 6–7) is interpretable as a confidence-weighted loss reduction, and token-level visualizations (Figure 5) provide intuitive validation (e.g., low scores for repetitive TikZ code).

**Weaknesses:**

* Computational cost: Scoring requires forward passes with both short and long contexts using a large model (Llama-3.1-8B), which may be prohibitive for very large corpora despite optimizations.

* Limited model scope: Evaluation is restricted to LLaMA-3-8B; generalizability to other architectures (e.g., Mamba, RWKV) or smaller models is unverified.

* Task coverage: Benchmarks focus on retrieval and structured reasoning; performance on narrative coherence or open-ended generation is not assessed.

* Baseline fairness: ProLong and LongFilter share high-quality data (since ProLong uses unfiltered data that includes LongFilter’s top samples), potentially underestimating LongFilter’s true advantage.

**Questions:**

* How sensitive is LongFilter to the choice of base model for scoring? Would a smaller or differently trained model yield comparable rankings?

* Could the scoring be approximated more cheaply (e.g., via attention patterns or n-gram statistics) without sacrificing effectiveness?

* Does LongFilter improve reasoning depth (e.g., multi-hop QA) or primarily retrieval fidelity? The current benchmarks emphasize the latter.

* How does performance scale with the selection ratio (e.g., top 10% vs. 20%)? Is there a point of diminishing returns?

---

> ### Author Response · Authors · 2025-11-29
>
> Thank you for your review and the valuable comments.
>
> > Computational cost & smaller models
>
> We added a new ablation study in Appendix D.1, where we utilize a 0.6B parameter model for data filtering. The results indicate that the data selected by this small model still yields performance gains when pre-training larger models. While we recommend using the strongest available model to maximize performance, this finding confirms that our method remains viable even under constrained computational budgets.
>
> Furthermore, data selection is a one-time investment. While the training stage is often repeated multiple times (e.g., for hyperparameter tuning, varying seeds, or architectural sweeps), the filtered dataset can be reused across all these runs. Therefore, the cost of selection is effectively amortized over the lifecycle of the project. Moreover, the scoring process requires only forward passes (no backpropagation), making it significantly faster and less memory-intensive than training steps.
>
>
> > Additional Tasks, such as long reasoning and long writing tasks
>
> We believe these tasks are not suitable benchmarks for evaluating this work because they depend heavily on post-training. LongFilter is a data selection method specifically designed for the **pre-training**stage.  To the best of our knowledge, most prior work on context extension[1], long-text data engineering[2], and even commercial LLMs[3] typically do not include this component at this stage of the pipeline.
>
> [1] Peng, B., Quesnelle, J., Fan, H., & Shippole, E. YaRN: Efficient Context Window Extension of Large Language Models. In The Twelfth International Conference on Learning Representations.
>
> [2] Fu, Y., Panda, R., Niu, X., Yue, X., Hajishirzi, H., Kim, Y., & Peng, H. Data Engineering for Scaling Language Models to 128K Context. In Forty-first International Conference on Machine Learning.
>
> [3] Liu, A., Feng, B., Xue, B., Wang, B., Wu, B., Lu, C., ... & Piao, Y. (2024). Deepseek-v3 technical report. arXiv preprint arXiv:2412.19437.
>
> > ProLong (Baseline) uses unfiltered data that includes LongFilter’s top samples, potentially underestimating LongFilter’s (our method) true advantage.
>
> We appreciate the reviewer's point that our baseline is "strong" because it includes the LongFilter samples. However, we respectfully maintain that the full dataset is the correct baseline. In practical pre-training scenarios, practitioners face a binary choice: utilize the entire corpus or apply a filter. Therefore, we must demonstrate that our selected subset outperforms the **default** strategy of using the whole corpus. If we were to compare against a baseline of only "rejected" data, it would not reflect a real-world use case. Our results show that pre-training on the LongFilter subset is superior to the default full dataset, which is a stronger conclusion than being superior to the rejected data.
>
> > Data Selection Ratio
>
> We have included experimental results in using different selection ratios in Appendix D. These results demonstrate that LongFilter is highly **robust** to variations in this parameter, maintaining consistent gains across a broad spectrum of ratios. We recommend prioritizing the quantity of long-context data: users should aim to train on as much long-distance-informative data as possible. Following standard practice in data selection, the exact filtering ratio should be determined by the user's available computing resources and total token budget.

---

### Author Response · Authors · 2025-11-29
**Paper Revision**

Dear reviewers,

We greatly appreciate your insightful comments and would like to express our gratitude for your valuable input. We have taken your feedback seriously and implemented changes to rectify the concerns that were brought up in the reviews.

1. We added experimental results of using a **smaller language model (0.6B parameters)** for data filtering in Appendix D. The results show that this smaller model can also successfully identify high-quality data. This experiment addresses some reviewers’ concerns about computational cost, as it demonstrates that we can use a lightweight model for filtering while still providing valuable data for pre-training larger language models. It also responds to several reviewers’ requests for additional experiments regarding the score model used in LongFilter.

2. We have included experimental results in using different **selection ratios** in Appendix D. These results demonstrate that LongFilter is highly **robust** to variations in this parameter, maintaining consistent gains across a broad spectrum of ratios. We recommend prioritizing the quantity of long-context data: users should aim to train on as much long-distance-informative data as possible. Following standard practice in data selection, the exact filtering ratio should be determined by the user's available computing resources and total token budget.

3. We clarified in the paper that the scope of this work is strictly limited to the **pre-training stage**. Since complex reasoning and open-ended generation capabilities primarily emerge during post-training, evaluating them is beyond the scope of this work. We have revised the manuscript to better define this boundary and manage reader expectations.

4. We added a discussion of the alternative formulation of the raw loss difference in Appendix E. The results demonstrate that our formulation provides a significantly higher fidelity approximation (lower error) to the KL divergence.

---

### Meta-Review · Area_Chair_6ED5 · 2026-01-07

**Summary:**

The submission proposes a data curation framework that selects long-context pretraining data based on the information gain provided by extended context.  It compares short (4k) contexts and long (64k) contexts.  The appropriate use of long contexts improves performance on a number of retrieval benchmarks.

**Reviewer Concerns:**

Reviewers were largely positive, but with some concerns about computational cost, the scope of evaluated model, and diversity of tasks.  The authors responded to the computational cost concern (see below), and explained the choice of evaluation setting.  While not perfect, these are positive responses to concerns.

**Reviewer Scores:**

3/4 reviewers initially gave a 6, while one gave a 4. The reviewer who gave a 4 focused in part on computational cost, which the authors responded to by using a cheaper data filtering network, showing that this setting still can give an improvement.  This is at least a partial improvement in paper presentation.  This same reviewer outlined a weakness that the benefits of the proposed method may be understated due to the evaluation methodology.

---

### Decision · Program_Chairs · 2026-01-26

Accept (Poster)